# An Evaluation of Medication Prescribing Patterns for Acute Migraine in the Emergency Department: A Scoping Review

**DOI:** 10.3390/jcm10061191

**Published:** 2021-03-12

**Authors:** Jun Hua Lim, Leila Karimi, Tissa Wijeratne

**Affiliations:** 1Department of Neurology, AIMSS, Sunshine Hospital, Western Health, Melbourne, St Albans, VIC 3021, Australia; Bowenlimjh@gmail.com; 2School of Psychology and Public Health, La Trobe University, Bundoora, VIC 3083, Australia; L.Karimi@latrobe.edu.au; 3Department of Medicine, Faculty of Medicine, Rajarata University, Saliyapura, Anuradhapuraya 50000, Sri Lanka

**Keywords:** migraine, acute care, emergency department, analgesic, triptan, opioid, NSAID

## Abstract

Migraine is one of the leading causes of disability worldwide and patients with acute migraine frequently present to emergency departments (ED). The current literature suggests that ED treatment of migraine headache varies across institutions. Considering this, we conducted a scoping review to summarize trends in medication prescribing patterns for acute migraine treatment in the ED setting. Trends were evaluated for factors influencing treatment choices, with particular attention placed on opioids and migraine specific therapy. This scoping review was based on the Arksey and O’Malley methodological framework and included studies published between 1 January 2000 and 31 May 2020. 14 publications met the inclusion criteria. The most common classes of medication prescribed were anti-emetics or Non-steroidal anti-inflammatory drugs (NSAID), but rates varied between studies. There was a concerning trend towards an underutilization of triptans and overutilization of opiates. The use of specific clinical treatment goals (e.g., two-hour pain free freedom response) was also not evident. Additionally, 88% (*n* = 8) of the nine studies commenting on adherence to hospital or evidence-based guidelines stated that practices were non-adherent. Overall, the reviewed literature reveals treatment practices for acute migraine in the ED are heterogeneous and deviate from established international recommendations.

## 1. Introduction

Migraine headache is a common, debilitating, and costly neurological disorder, affects up to 16.6% of the general population [1] and is recognized as the second most disabling condition in the world according to disability adjusted life years [2]. Beyond individual wellbeing and suffering, migraine is a public health issue that puts a burden on society through healthcare system and productivity costs. For example, in 2018 migraine cost the Australian economy $35.7 billion AUD in direct and indirect costs [3]. Global studies of the burden of migraine have reported similarly troubling numbers in Europe and the US [4,5,6].

Migraine sufferers often present to the emergency department (ED) seeking relief from their symptoms, with data from the United states showing at least 1.2 million presentations to the emergency department every year [7]. When compared with other health services EDs received almost 20% of migraine presentations [8]. These patients pose a treatment challenge for EDs as they sometimes have severe and/or prolonged not typical of their usual headache; and/or have tried their usual migraine treatment without success [9]. To add to the complexity, the available treatments for migraine are varied and may include paracetamol, antiemetics, nonsteroidal anti-inflammatory drugs (NSAIDs) as well as migraine specific therapies (ergots and triptans) and opiates. The complexity of migraine as a neurological disease is one of many reasons for heterogeneity in acute migraine treatment [10,11].

Evidence based recommendations for acute treatment of migraine headache in ED offer guidance and proposals for streamlined management. Despite this, innumerable studies show deviation from these guidelines. Emergency departments typically adopt treatment choices conflicting with evidence-based guidelines, sometimes choosing suboptimal pharmacological treatment. For example, the practice of prescribing opiates for migraine is well documented in the literature despite its association with chronification of migraine, development of medication overuse headache [12] and its propensity to harm patients through withdrawal or dependence [13,14]. Cross-institutional information regarding ED physician treatment practices for migraine is not well described in the current literature, and the amount of information currently available is unknown. For this reason, a scoping review was performed to map the current medication prescribing patterns and elucidate any gaps in knowledge in this area.

The current scoping review sought to answer the question: what are physician preferences for prescribing acute medication for the treatment of migraine in the ED? The scoping review was further guided by the following questions:What classes of medication were most frequently prescribed for the acute treatment of acute migraine in the ED?What are the rates at which narcotic and migraine specific medications are prescribed in the ED?What factors influenced preferred treatment of migraine in the ED?What factors precluded adherence to evidence-based migraine guidelines?How can migraine treatment in ED be more consistent with current evidence-based guidelines?

## 2. Experimental Section

### 2.1. Eligibility Criteria

#### 2.1.1. Types of Participants

The scoping review will consider any published studies on any patients over 18 presenting with migraine who presented to any emergency department. Studies in pregnant populations were excluded due to a lack of generalizability.

#### 2.1.2. Concept

The concept of interest was the prescribing patterns for the treatment of acute migraine in the ED. To be included in this review papers needed to focus on describing patterns of medications used for migraine patients in the emergency department.

#### 2.1.3. Types of Study

The types of study considered for this review included case control studies, cohort studies, randomized controlled trials, systematic reviews and meta-analyses. All peer reviewed experimental, descriptive and observational studies reporting quantitative data were considered.

We considered studies which aimed to quantitatively describe patterns of medication use for migraine patients. Migraine patients included those who were diagnosed with migraine headache by a medical professional or were determined to have migraine based on the either the International Classification of Headache Disorders (ICHD) or International Classification of Diseases (ICD) criteria. Studies including those who were self-diagnosed with migraine or were not migraine specific (e.g., studied all primary headaches) were excluded.

### 2.2. Methods

Peer-reviewed journal papers were included if they were relevant to the concept mentioned above, were published between 1 January 2000 and 31 May 2020, written in English and involved human subjects who were non-pregnant adults. This scoping review was conducted based on the Arksey and O’Malley methodological framework [15], and to identify all relevant publications for the current review the following three-step search strategy was used:A search of MEDLINE, Cochrane, and the Cumulative Index to Nursing and Allied Health Literature (CINAHL) was conducted, followed by an analysis of words contained in the title and abstract, as well as index terms used to describe relevant articles.All identified keywords and index terms were used to conduct a second search using the following databases: Ovid MEDLINE, Cochrane Library/Systematic Reviews, PubMed, CINAHL, PsycINFO and Excerpta Medica database (EMBASE).A manual search as conducted to ensure all relevant studies were included.

The following keywords were used with the closest corresponding relevant subject headings: migraine, acute care, emergency department, analgesic, triptan, opioid and NSAID. The search strategy was performed by the authors in conjunction with advice from an experienced librarian.

### 2.3. Screening and Data Extraction

The final search strategy results were exported into Endnote X9. The web-based tool Covidence was used to aid the process of removing duplications, screening, and data extraction.

The data screening process was jointly developed by the reviewers. A consensus was reached on which variables to extract and the extraction of data was performed by one reviewer. It was then reviewed and agreed upon by another reviewer in situations of doubt.

Data from the included studies was extracted to address the review questions using the methods outlined by Peters et al. [16]. The extracted data included: basic article characteristics (e.g., author, year of publication and title), demographic information (subject population, age and gender), study type (e.g., retrospective or prospective), main aim and outcomes, method for determining inclusion (e.g., use of International Classification of Diseases (ICD) coding, International Headache Society Classification (IHSC), or physician diagnosis of migraine, medications described (i.e., whether the study aimed to describe general prescription patterns or focused on comparing groups such as opioid vs. non-opioid treatment), key findings (focusing on the outcomes of the present review) and additional observations (focusing on other relevant outcomes not identified by this study’s research focus). Some items extracted required interpretation by the reviewer, for instance in the outcomes section reviewers may have disagreed on what outcomes were relevant to the main research question.

## 3. Results

Our search strategy yielded 1302 studies (a breakdown of pooled results is available in Appendix A), and 818 remained for screening after duplicates were removed. The 818 studies underwent title and abstract screening. Following this, 252 full text studies were assessed for eligibility. Of these studies, 267 were excluded due to: inappropriate outcomes (prescribing patterns were not described), inappropriate study population, inappropriate study design, inappropriate setting, and not meeting the inclusion criteria (refer to Appendix A for full details). This process resulted in 14 studies which were selected for inclusion in the current review. Figure 1 shows a Preferred Reporting Items for Systematic Reviews and Meta-Analyses (PRISMA) diagram depicting the publication screening and selection process (for elaboration on pooled numbers please refer to Appendix A).

**Figure 1 jcm-10-01191-f001:**
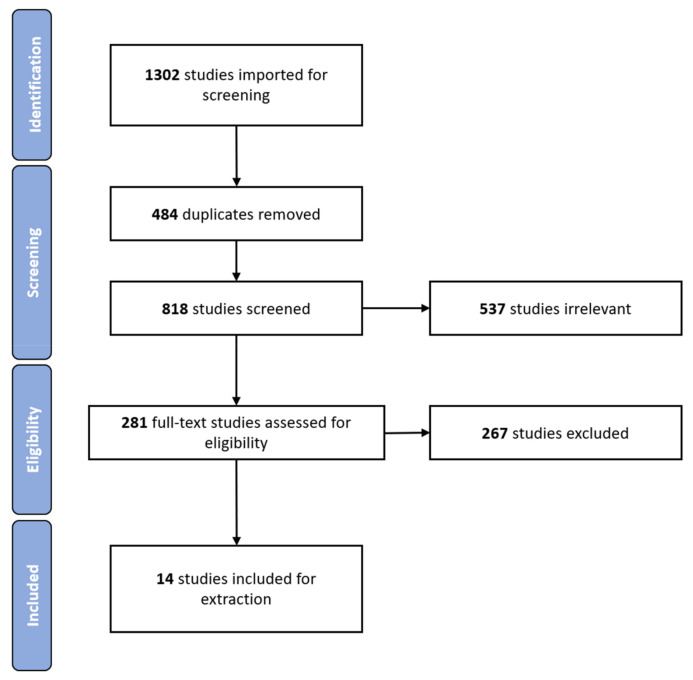
PRISMA flowchart indicating the publication screening and selection process.

**Table 1 jcm-10-01191-t001:** Characteristics of included studies.

**Author, Year**	**Country and Setting**	**Main Aim of the Study**	**Study Type**	**Study Size**	**Age (Mean Years, Unless Stated Otherwise)**	**Gender**	**Migraine Population and Definition**	**Medication**
Gunasekera, 2020 [17]	Australia: St Vincent’s Hospital, Melbourne	To determine whether the emergency department (ED) prescribed medications were consistent with national guidelines.	Retrospective cohort study	744 patients	36.4	M 25%, F 75%	Patients admitted to St Vincent’s Hospital Migraine diagnosis based on the G439 international Classification of Disease, tenth revision (ICD-10) code.	Opioids and generalized prescribing patterns
Minen, 2020 [18]	United States: 2 urgent care locations, New York	To examine the treatment and management of migraine patients admitted in the ED, focusing on discrepancies between prescribed therapies and the American Headache Society (AHS) migraine management guidelines.	Retrospective chart review	78 patients	32.5	M 20.5%, F 79.5%	Patients New York University (NYU) Langone Manhattan Urgent Care center or NYU Langone Ambulatory Care West side Urgent care Centre Migraine diagnosis made by a physician	Generalized prescribing patterns
Shao, 2020 [19]	United States: Baylor Scott & White Health services, Texas	To describe the use of opioid and non-opioid medication in patients admitted to ED with migraine and compare demographics between opioid users and non-opioid users in the same group.	Retrospective study using electronic health records	12,945 patients 788 met the inclusion criteria	44.5	M 14.1%, F 85.9%	Patients enrolled for 6 months before or after their ED visit. Migraine diagnosis based on International Classification of Disease, ninth revision (ICD-9): 346.xx or ICD-10: G43.xxx.	Opioids and generalized prescribing patterns.
Ruzek, 2019 [20]	United states: 4 suburban EDs, New Jersey	To determine how migraine treatment in the ED has changed between the years 1999–2000 and 2014, with a secondary goal of ascertaining if there was a change in the return rate to ED in 72 h over the years.	Retrospective cohort study	8046 patients between 1999–2014 (72-h return group) 290 chart reviews (147 in 1999–2000 and 143 in 2014)	Chart review: 38 72-h return group: 38	M 11%, F 89% (chart review) M 16%, F 84% (72-h return group)	Migraine diagnosis based on an ED physician diagnosis of migraine coded with the respective ICD-9 code	Generalized prescribing patterns
Shao, 2017 [1]	Australia: A large metropolitan ED, Queensland	To identify the varying demographics of patients which presented to the ED with a migraine and analyze the trends in medication treatments and prescriptions for the migraine.	Retrospective study of clinical records	2228 patients	Migraine patients: 37.05ED population: 46.17	M 29%, F 71%	Migraine diagnosis required both An ICD code primary of G43.9- migraine; and a presentation to ED using the word “migraine” as a description as a primary complaint.	Generalized prescribing patterns.
Young, 2017 [21]	United states: 3 emergency departments, Connecticut	To explore and describe treatment habits in three different settings: an academic medical center, a non-academic urban ED, and a community ED	Retrospective cross-sectional analysis of consecutive adult emergency visits	1222 unique Visits 931 unique patients	36 (Median)	M 17.3%, F 82.7%	Migraine diagnosis based on ICD-9 code for migraine or one of tis variations ICD-9 code 346 or 346	Opioids and generalized prescribing patterns
Berberian, 2016 [22]	United States: An academic emergency department, Pennsylvania	To determine the frequency with which parenteral narcotic analgesia is used to treat acute migraine in an academic ED and to compare the cost and length of stay between patients treated with narcotic vs. non-narcotic treatments	Retrospective cohort study	421 subjects treated with parenteral agents. From a total of 521 records of patients diagnosed with migraine	Not stated	Not stated	Migraine diagnosis based on both chief complaint and ICD-9 codes for Migraine taken from electronic records	Opioids prescribing patterns
Cheng, 2016 [23]	Australia: 2 metropolitan hospital EDs, Melbourne	To analyze the demographics, presentation, management, and outcomes of patients who presented to the ED with a migraine, making a comparison between first presenters and those with a history of migraine.	Retrospective cohort study	356 patients	37.8	M 25.2%, F 74.8%	Migraine diagnosis defined by fulfilling a headache which had no organic cause a discharge diagnosis of migraine headache ; a presentation to ED complaining of “migraine”	Generalized prescribing patterns
Friedman, 2015 [7]	United states: Randomly selected EDs, multiple sites across the US	To make a comparison between the frequency of current medications given to patients with acute migraines in EDs within the United States (US) with those used in 1998. The authors also aimed to identify factors related to the use of opioids	Retrospective study using 2010 data from the Hospital Ambulatory Medical Care Survey (NHAMCS)	1.2 million visits to ED for migraine	Not stated	Not stated	Patients with a ICD-9 coded discharge diagnosis of migraine	Generalized prescribing patterns
Supapol, 2013 [24]	Canada: 12 emergency departments, Ontario	To evaluate the prevalence of opioid therapy as a primary treatment for migraine headache in 12 Ontario EDs by randomly selecting 100 migraine patient charts	Retrospective study	100 randomly selected patient charts	not stated	Not stated	Migraine diagnosis defined though clinical charts with a National Ambulatory Care Reporting System (NACRS) most responsible diagnosis (MRDX) coding of migraine	Opioid prevalence
Valade, 2011 [25]	France: 20 general emergency departments, multiple sites across France	To determine the proportion of headache patients diagnosed with migraine, and to ascertain demographic and clinical characteristics of these patients and describe the treatment and follow-up they received.	Prospective observational study	15, 835 patients were admitted to ED 483 (3.1%) had a headache. 98 (0.6%) had migraine.	37.6	M 25.5%, F 74.5%	Migraine diagnosis based on a questionnaire completed by an ed physician containing: The International headache Society (IHS) criteria Disease history and treatment The reason for the emergency department visits whether further examination was required treatment prescribed including prescription medicines at discharge.	General prescribing patterns
Tornabene, 2009 [26]	United States: 2 emergency departments, California	To examine and compare the treatment type and throughput times of migraine patients between an urban and suburban ED, and between patients that visited the ED multiple times (repeaters) vs. only once (non-repeaters).	Retrospective review of patient records	189 patients 249 total visits	Repeaters: 40.9 Non-repeaters: 39.5	Repeaters: M 36.2%, F 63.8% Non-repeater: M 24.2%, F 75.8%	Based on 2 criteria: ICD-9 classification and ED diagnosis of “migraine headache” or “migraine” as determined by a physician	Opioids versus non-opioid prescribing patterns
Wasay, 2006 [27]	Pakistan: An emergency department, Karachi	To discern whether Internation Headache Society (IHS) guidelines were being met within a tertiary care hospital ED in Pakistan.	Retrospective cohort study	161 patients	34	M 36%, F 64%	Migraine diagnosis based on the IHSC criteria	Opioids versus non-opioid prescribing patterns
Freidman, 2009 [28]	United States: 2 Emergency departments, New York	To determine the proportion of migraine patients presenting to ED who were treated with migraine-specific therapy as well as to note the amount of unnecessary neuroimaging studies performed.	Retrospective cohort study	156 patients	Not stated	M 19.8%, F 80.2%	Migraine diagnosis based on patients with the ICD-9 codes 346.0, 346.1, or 346.9 and a primary diagnosis of migraine	Migraine specific versus non migraine specific therapy

Abbreviations: AHS American Headache Society, ED Emergency department, ICD International Classification of Diseases (IDC-9 for ninth revision and-ICD-10 for tenth revision), IHSC International Headache Society Classification ,NYU New York University, US United States.

A table of study characteristics from the extracted articles is provided in Table 1. Table 2 describes the main findings and additional observations (insights relevant to the current review but not directly addressed in the aims).

The 14 Included publications were published between 2009 and 2020 with most studies from developed countries. This included: 57% (*n* = 8) from the United States, 21% (*n* = three) from Australia and 7% (*n* = one) from Pakistan, France, and Canada. The study population with migraine ranged between 78 [18] and 2228 [1] subjects. Most migraineurs presenting to ED were young and female with average age ranges between 32.5 [18] and 44.5 [19] years and females consisting between 64% [27] and 85.9% of patients [19].

Of the 14 included publications all used observational data, and all but one of the publications were retrospective [25]. Whilst each study made comments on prescribing patterns observed, the medication classes which they primarily focused on were different. About half (57%) of the publications focused on the use of opiates [17,19,21,22,24,26,27], about a third (35%) sought to describe general prescribing patterns [1,18,20,23,25,29] without comparing specific classes of medications and only one study compared the use of migraine specific therapy to non-specific therapy [28].

Overall antiemetics as a general class were reported as the most used medication in 50% of studies [1,17,19,20,21,22,28], followed by non-opioid analgesics at 28.5% [18,23,25,27]. Intravenous fluids were the most commonly were reported therapy in 14% [1,20] of studies and 21% studies did not include detailed descriptions of non-opiate prescription patterns [24,26,29]. Regardless of the study context comparisons of different pharmacological treatments against opioid therapy were frequently made. Overall, there is a clear trend towards opiate overutilization with 12 studies observing a prescription rate of over 10 [3,5,8,22,23,24,26,27,28,30,31,32]. Moreover some studies showed very high rates of opiate prescription, in one hospital the prescription rate was 69% [24]. Interestingly, all extracted studies commenting on the consequences of patients who were prescribed opiates described poorer outcomes [21,22,26] or no increased pain relief [27]. Interestingly none of the studies mentioned concepts such as sustained pain free response or 2-h pain freedom as a treatment goal, both of which are important clinical endpoint measures which can be used to assess the outcomes of migraine treatment [33]. This suggests that opiates are being overprescribed to patients with migraine without any clear clinical goals of treatment.

From our extracted studies, we see that the prescription of opiates to migraine patients has been a long-standing issue that has been described in the literature for at least the last 10 years. Whist the trend of over prescription is clear the factors contributing to this trend are unclear. There was no clear consensus between studies commenting on the influence of demographic factors on opiate prescription rates [1,7,17,19,22]. A study examining nationwide data in the US, showed no relationship between region and opiate prescription patterns [7]. This observation may explain large variations of prescription patterns within a single region, for example 6.9% to 69.9% in one study [21] and 0% to 69.9% [24] in another. 

Several of the extracted studies have stated a lack awareness as a primary contributor to high rates of opiate prescription for migraine headache [17,21,25,27,30]. Specifically, a lack of awareness or familiarity or knowledge of alternative medications or the harms which opiates pose to migraineurs. An acknowledgement of the importance of education and awareness was expressed in 5 of the extracted studies [17,21,25,27,30]. Regarding triptan prescription practices, of the extracted studies two noted that no parenteral triptans were used as they were not stocked in the pharmacy, and an additional two studies noted that triptan prescription was under 1%. The highest rate of triptan use was noted by Freidman et al. (2009), at 11.5% (*n* = 9). Of the extracted studies describing triptan prescribing, only Freidman et al. (2009) mentioned the proportion of eligible patients receiving triptans.

This raises the issue of non-compliance with evidence-based guidelines whether from the AHS or otherwise. Indeed, of the nine studies which commented on adherence to hospital or clinical practice guidelines, 88% stated that their practices were non-adherent. The one study which commented that their ED practices were within the American headache society guidelines mentioned that there was room for improvement, as they used opioid analgesics for migraine 24% of the time.

## 4. Discussion

The current review attempted to characterize the treatment of migraine headache in ED with a particular focus on opiate and triptan prescription patterns, rationalizations for medication choice and lack of adherence to evidence-based guidelines. The demographics in the included literature were consistent with the published literature literature [31,32,34,35] where higher numbers of female migraineurs at younger ages were reported. Furthermore, our results revealed that anti-emetics and non-opioid analgesics (e.g., paracetamol or NSAIDs) were the most prescribed medications for migraines in ED. We also found that opiates continue to be overprescribed despite evidence of poor clinical outcomes. This occurs without any clearly defined clinical goals of treatment. A similar issue exists regarding triptan prescription. Subcutaneous sumatriptan is recommended by the American headache society guidelines [10] as a number of randomized controlled trials and systematic reviews have found that the triptan class are effective in the treatment of acute migraine [36]. Despite this recommendation, triptans were under prescribed. However, our results suggest the trend is potentially over reported due to lack of eligibility reporting.

The variation in opiate and triptan prescription rates suggests a lack of consistency in the ED approach to acute migraine management. More needs to be done to reduce opiate prescription and increase triptan use for eligible patients. Not only will these goals relieve the suffering of patients, but the extracted studies also suggest that reducing opiate treatment may lessen the length of patient stays or representation to ED.

While the trend of opioid over prescription and triptan under prescription is clear, the extent of non-compliance with evidence-based medicine is unclear. To characterize the extent of opiate prescription, acceptable opiate use in accordance with AHS guidelines must be separated from unacceptable use of opiates. The AHS guidelines classifies the use of intravenous opiates as ‘May avoid- Level C’ [10] and the American Academy of Neurology acknowledges that opioids are considered rescue therapy but should be used infrequently [12]. It is believed that a proper characterization of whether opiates are used as first line or rescue therapy is needed to make a complete judgement on the appropriate use of opiates within an institution. It is our opinion that to comply with best practice guidelines on opiates, an ED should employ near zero levels of opiate as first-line therapy for migraine and opiates should not encompass the majority of rescue therapy prescriptions. We encourage future research to consider these details when conducting observational studies for opiate prescriptions in the ED.

The extent of triptan non-compliance is also unclear. Evidence is highest for subcutaneous sumatriptan within AHS guidelines, along with intravenous prochlorperazine and metoclopramide: “should offer—level B”. As mentioned previously, the proportion of triptan-eligible patients is not frequently reported. We suggest that those conducting observational studies on triptans versus other non-migraine specific treatments consider separating triptan-eligible patients from ineligible patients to properly assess adherence. Implementing these measures may reduce the amount of triptans we can reasonably expect EDs to prescribe in order to be consistent with AHS guidelines.

The extracted papers rationalized non-adherence in different ways, the most consistent rationalization for opiate overuse being a lack of awareness of guidelines as supported by five of the extracted articles [17,18,21,25,27]. This phenomenon can rationalize the variation in prescription rates within a limited geographic area. For example, the comparison between an Academic center, Urban ED and community ED by Young et al. (2017) found that the academic ED had the lowest prevalence of opiate prescription. Here the authors suggested ineffective knowledge translation of evidence-based migraine guidelines as a potential cause for difference in prescribing practices between hospitals. This is because the academic center was the only location with residents and teaching facilities. For this reason, educational interventions may help alleviate this issue. There was limited evidence on the efficacy of educational interventions; however, Wasay et al. (2006) acknowledged that their relative opiate prescription rates were potentially lowered by a workshop organized by the local neurology department.

Rationalizations for low rates of triptan use were less consistent. Shao et al. (2020) suggested evidence that their reduced effectiveness in patients with late attacks may play a role. Additionally, given that sumatriptan is contraindicated in common conditions including cardiovascular disease and pregnancy, the appropriateness of its use may be narrower than we previously thought. Shao et al. (2020) also believed that frequent occurrences of adverse effects in 50% of patients after triptan administration may lead physicians to avoid use. Comments made by Young et al. (2017) were also valid, suggesting that individual factors such as previously reported poor response to triptans by patients, physician unfamiliarity with medication and high cost of triptans may also contribute to low use. The characterization of such details in future studies will further elucidate the extent to which triptan prescription practices deviate from guidelines, potentially allowing a more targeted solution to this problem.

Finally, the last crucial issue in addressing variation from guidelines is the lack of availability of triptan medication. Wasay et al. (2006), based in Pakistan, reported that subcutaneous triptans were not available in Pakistan. Similarly, Minen et al. (2020), based in New York USA, reported that the emergency department pharmacy does not stock subcutaneous sumatriptan or intravenous prochlorperazine, both of which are recommended by the AHS guidelines.

Overall, we believe that the simplest intervention to address the tendency for ED overutilization of opiates and underutilization of triptans is to increase awareness of evidence-based guidelines through educational interventions whilst also increasing the availability of medications recommended. Moreover, we believe future observational research into this area should focus on further characterizing how prescription patterns deviate from recommended guidelines, specifically, whether opiates are being prescribed as first line treatment or rescue therapy and noting the eligibility of patient populations when commenting on levels of triptan therapy. Otherwise, we believe that a prospective study on the effect of education regarding guidelines may be beneficial for improving migraine treatment in ED.

### Limitations

The current study only focused on quantitative studies where the primary aim was to describe medication prescription patterns. Our original objectives focusing on the rationale for, or factors precluding adherence to, evidence-based guidelines were not directly addressed in many papers. While we feel that our discussions are likely to be based on widely held beliefs in the field, due to the scope of our review there is a possibility that the opinions gathered may not be comprehensive. Furthermore, limiting the search to quantitative research may have excluded articles covering the qualitative aspects of non-adherence to recommended guidelines. Finally, we may not have a global view of the issue, as many articles used in the review are from very highly developed countries. The ability to gain a global perspective on the issue is further hindered by only reviewing literature published in English. While a global perspective may not have been achieved, it is felt that the narrative provided here will be beneficial and translatable to many other contexts.

## 5. Conclusions

Overall, the current scoping review shows that there is a worrying tendency towards the over prescription of opiates and under prescription of triptans. We observed a significant evidence–practice gap in the management of acute migraine in ED. The published papers continue to support the notion that migraine headache continues to be the most neglected, worst respected, worst managed medical disorder in the world [37]. Ongoing advocacy, educational programs and translational research in this area should focus on addressing this issue as a matter of high priority.

## Figures and Tables

**Table 2 jcm-10-01191-t002:** Key findings and additional observations of included studies.

**Author**	**Key Findings**	**Additional Observations**
Gunasekera, 2020 [17]	Migraine management in ED was inconsistent with national guidelines Opiates were overused as 46.4% (*n* = 325) of patients received them Triptans were underused as only 6.9% (*n* = 51) of patients received them 60% (*n* = 451) of patients were given antiemetics, 51.8%(*n* = 385) received paracetamol and 37% (*n* = 274) where given Non-steroidal anti-inflammatory drugs (NSAIDs)	The most common reason for presenting to the ED with migraine was failed treatment at home (*n* = 480, 64.5%). Both migraine-related and non-migraine-related previous opioid use, previous ED visits, age, and diagnosis of sleep disorder might help identify migraine patients with high risk of opioid use. The authors state multiple reasons for high rates of opiate prescriptions including: ED doctors ‘treating on the run’ (i.e., treating pain without patient review), a lack of education regarding management of acute migraines, or barriers to implementing migraine treatment.
Minen, 2020 [18]	93.6% (*n* = 73) of migraine patients had pain in ED. Only 12.3% (*n* = 9) received acute migraine treatment medication within the ED consistent with AHS guidelines Of these patients 46.6% received no medication. Ketorolac was the most common choice of medication (70.6%). No patients received subcutaneous sumatriptan or intravenous (IV) prochlorperazine. 25.6% (*n* = 20) received a prescription for triptans	Of the AHS recommended treatments (IV metoclopramide, IV prochlorperazine, and subcutaneous sumatriptan) only IV metoclopramide was found in the pharmacy In their publication the authors suggested several possible explanations for high opioid use
Shao, 2020 [19]	35.9% (*n* = 283) of patients received opiates 0.4% of patients received triptans in ED and 0.5% of patients were given triptans on discharge Other common classes of drugs included: antiemetics (*n* = 292, 37.1%), nonopioid analgesics (*n* = 246, 31.2%), antihistamines (*n* = 153, 19.4%), and corticosteroids (*n* = 74, 9.4%). Adherence to the guidelines was minimal given the low triptan use and a high rate of opiate prescription (15%)	After controlling for covariates, several predictors of index date opioid use were found this included: previous migraine-related opioid use (odds ratio 1.66–4.43) non-migraine-related opioid use (10 or more prescriptions, Odds Ratio: 1.93) previous all-cause ED visits (1–3 visits, Odds Ratio: 1.84) age (45–64 years, Odds Ratio:1.45) patients with sleep disorder(Odds Ratio:1.43).
Ruzek, 2019 [20]	The use of IV fluids (88%), Dopamine receptor antagonist (83%), ketorolac(38%), and dexamethasone(22%) was more common in the 2014 cohort compared to 1999–2000 cohort. Narcotic prescriptions in ED and on discharge was lower in the 2014 cohort compared to 1999–2000 cohort.	Authors acknowledge that hospitals in their area or in other geographic regions may have different treatment practices for migraines. 8% (*n* = 624) of all migraine patients between 1999- 2014 represented after 72h; this return rate was lower in 2014 (4%) compared to 1999–2000(12%) The authors speculate that the increased use of non-narcotic medications contributed to the decrease.
Shao, 2017 [1]	A variety of medications were used to treat migraine. Simple analgesics, anti-emetics and IV fluids with phenothiazine were commonly used. 20% of patients received oral or parenteral opiates (42 of 194 initial medication prescriptions, and 64 of 292 as required medication prescriptions). Use of metoclopramide and phenothiazines were commonly prescribed for migraines and are consistent with National Health and Medical Research Council (NHMRC) guidelines. Despite the triptans being recommended in the guidelines, their use was minimal.	The authors found that the proportion of ED patients presenting with migraine is steadily increasing. Additionally, the migraine population was trending towards younger patients (Mean = 37.05, standard deviation (SD) = 13.23) than the whole ED population (Mean = 46.17 SD = 20.50) (*p* < 0.001). ∙
Young, 2017 [21]	Opioids were given for migraine headache in 35.8% of the 1222 ED visits. Opioid prescription rates varied between centers and ranged from 6.9%–69.9% depending on center and rescue vs. first-line therapy. The most common first line treatments included: antiemetics at 35.3% of all orders, NSAIDs at 16.0%, IV fluids at 13.3%, and opioids at 12.6%. Triptans were given to 1% of patients(*n* = 2).	Patients receiving opiates had a 37.7% increase in their length of stay compared to non-opioid treated counterparts [95% Confidence interval (CI) 1.207 to 1.617]). In comparing three facilities (An academic center, urban ED and community ED) the community ED had the highest prevalence of opiate prescription. The academic center was the only location with residents and teaching facilities. Hence, the authors suggested there was evidence of ineffective knowledge translation of evidence-based migraine guidelines creating a difference in prescribing practices.
Berberian, 2016 [22]	32% (*n* = 134) of patients were treated with one or more parenteral narcotic analgesic agents. The most used medications were antiemetics (92%), antihistamines (70%) and non-steroidal anti-inflammatory medications (62%).	The use of parenteral narcotics was associated with a longer length of stay (mean time 5:03 vs. 4:06 min, *p* = 0.001) but a reduced cost of say (mean cost $2363.62 vs. $4528.82, *p* = 0.00008) The authors noted there were no significant age, gender, or race differences among those administered narcotics versus those not administered narcotics.
Cheng, 2016 [23]	The types of management in the ED were varied. However, except for the use of IV fluids and parenteral dopamine antagonists, migraine management was similar for patients with a previous history and first presenters. The initial migraine management included paracetamol in 48.8% of cases (*n* = 178), NSAIDs in 51.2% of cases (*n* = 187). Opioids were given in 25.8% of cases (*n* = 94) Triptans were used in 12.6% of cases (*n* = 46) and ergots were used in 0.5% of cases (*n* = 2). A significant proportion of patients with a migraine history received parenteral dopamine antagonists (62.5%)	The median length of stay in the ED was 4 h, with 163 (44.7%) patients admitted to the short-stay unit. Those that required a short stay in the ED were often discharged from the hospital despite persistent pain.
Friedman, 2015 [7]	In 2010, opioids were administered in 49% of ED visits (95%CI: 51, 67%). Choices of opiates have changed when comparing 2010 prescriber preferences those in 1998. In 2010, 77% (95% CI 70, 83)) of migraine patients were given anti-emetics.	Female patients received opioids more often than male patients Opioids are less frequently given to patients who have never visited the ED (compared to those that visited the ED 1–3 times per year Opioids were commonly used for migraine regardless of demographic features, region of the country (Northeast, Mid-west, South, West), type of hospital (non-profit, Government, or proprietary) or presenting level of pain. The authors hypothesized several reasons emergency clinicians might have chosen opioids to treat migraine headache
Supapol, 2013 [24]	Large variations in opioid prescription rates were observed: this ranged from 0% (95% CI: 0–3.0) to 69% (95% CI: 59–77) at 6 of the 12 hospitals. 6 of the 12 hospitals were adherent to the standard which was a 0–8% opiate prescription rate. The other 6 had proportions ranging from 24% to 69%.	At two hospitals where more than 60% of patients received opiates, a small number of patients were responsible for 45% of migraine visits. Half of the hospitals in the LHIN met the standard for opioid primary therapy; 5% or less primary opioid therapy is an achievable goal.
Valade, 2011 [25]	Fewer than expected patients received migraine-specific prophylactic treatment The most common treatment pharmacological treatment prescribed were non-opioid analgesics (61.2%) and NSAIDs (42.9%) respectively. Opiates 5%(*n* = 1) of 20 migraine with aura patients received weak opiates, where as 9.1% (*n* = 7 ) of 77 patients with migraine without aura received weak opiates. Triptans were given less often (11.2%). 9% of patients received no treatment.	Upon discharge most patients (80.2% of 92 patients) did not experience a resolution of their migraine. Patients often received non-specific medication upon discharge which failed to resolve symptoms. Follow up data showed 36.3% of patients did not have complete resolution of their migraine after 48 h. Their results suggest potential for improved treatment choice, and that awareness of guidelines is needed
Tornabene, 2009 [26]	68% of migraine patients in ED were treated with opioids. Within this group 81% were given opioids as their initial pharmacological treatment, and 38% received multiple doses.	Patients who were prescribed opioids stayed in the ED significantly longer that those with non-opiate treatments: 142 min (95% CI 124–160) vs. 111 min (95% CI 93–129) (*p* = 0.015), respectively Repeat patients who visited the ED multiple times were more likely to be treated with opioids compared to non-repeaters, 90.6% (*n* = 87) vs. 54.2% (*n* = 83), respectively. Repeat visitors were more likely to be given multiple doses of opiates (41.6%, *n* = 40) compared to 15.7% of non-repeaters (*n* = 24)
Wasay, 2006[27]	This study found that opioid analgesics were used as first line migraine therapy in 24% of patients. The remaining patients (76%) were treated with non-opioid analgesics. Parenteral triptans were not available in this hospital setting however overall, it concluded due to circumstances and a low rate of opioid use that the Aga Khan University ER did seem to be within the guidelines of the Headache Society. The authors acknowledge there is potential for further improvement. The authors state that “Only two oral triptans are available in Pakistan: sumatriptan and zolmitriptan; but awareness regarding the usefulness of these medications for acute migraine is limited to neurologists”	100 (62%) patients were discharged with full relief of pain 50 (31%) patients had partially relieved pain, 11 (7%) patients in pain upon discharge. The study found no relationship between pain relief and the prescription of opioid vs. non-opioid therapy
Freidman,2009 [28]	Migraine-specific treatment is underutilized in the ED: Of 156 patients the majority were treated with various parenteral antiemetics(*n* = 95), narcotics (parenteral opioids, *n* = 63; oral opioids *n* = 19), or ketorolac (*n* = 65). 78 patients (50%) had potential contraindications to receive migraine-specific therapy, hence justifying non-specific therapy. Of the 78 patients eligible for migraine-specific therapy, only 9 (11.5%) were able to receive migraine-specific therapy, while 10 patients received no treatment at all. Overall migraine specific therapy was underutilized, however patient eligibility for such therapy may explain this.	The cost of radiological investigations was a major contributing factor to the overall financial burden of emergency care for migraine patients.

Abbreviations: AHS, American Headache Society, CI Confidence interval, ED Emergency department, ICD International Classification of Diseases (IDC-9 for ninth revision and ICD-10 for tenth revision), IHSC International Headache Society Classification, IV Intravenous, NSAID Non-steroidal anti-inflammatory drug, SD Standard deviation, NHMRC National Health and Medical Research Council.

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
