# Peer review of "An Evaluation of Medication Prescribing Patterns for Acute Migraine in the Emergency Department: A Scoping Review"

_jcm, 2021, doi:10.3390/jcm10061191_

Round 1
Reviewer 1 Report
The authors performed a scoping review to shed light on several important aspects regarding the ED management of migraine. The article requires significant changes, including modifications that will render it more reader-friendly. Methodological, reporting issues and inaccuracies are present, while several conclusions are inconsistent with the study findings (some examples are given below). Region and time trends, as well as factors associated with treatment choices, are insufficiently commented on.
Comments per section:
Abstract: I suggest that you remove all references from the section Abstract.
Introduction: 1st Paragraph: 2nd Sentence: The word ‘It’ refers to ‘migraine headaches’ (1st sentence). Please change one of these terms.
Introduction: 2nd Paragraph: 1st Sentence: ‘and the ED is thought to provide 20% of migraine care’, please be more specific (in terms of cost? disease burden?).
Introduction: Last Paragraph: This paragraph, summarizing your findings, should be removed from the section Introduction.
Experimental Section: Types of studies: Please provide the types of eligible studies more clearly. ‘Peer-reviewed quantitative, mixed-methods’ is not a type of study. ‘Cochrane systematic reviews’ in particular? What about other reviews? ‘Case studies’ do not reflect the prescribing patterns; therefore, they should not be eligible.
Experimental Section: Methods: A scoping review should be based on a comprehensive and repeatable literature search. Although your methodology is correct, your description does not ensure repeatability. I suggest that an exact description is provided in the context of ‘supplemental material’. Transparent reporting of the literature search is important to facilitate the future performance of systematic reviews. For example, there is no description of the ‘manual search’ (references of the retrieved articles? studies citing the retrieved articles?). ‘All identified keywords and index terms were used to conduct a second search’, sharing these terms would facilitate future literature searches. How come both MEDLINE and PubMed were searched, considering their overlap? Please modify this section.
Experimental Section: Screening and Data Extraction: ‘The screening the web-based tool Covidence’, please reformulate.
Results: Please provide the number of studies retrieved from each individual database, not only pooled numbers. Please present the number of studies corresponding to the individual reasons for exclusion.
Results: Data screening and Results: Please move the methodological information (data screening process, nature of extracted data) to the section: Experimental Section: Screening and Data Extraction. Also, the methods for determination of inclusion (specifically: use of international classification of diseases (ICD) coding and physician diagnosis of migraine) are broadened in comparison with the eligibility criteria you used: ‘Migraine patients included those who were diagnosed with Migraine headache by a medical professional or based on the International classification of headache disorders (ICHD) migraine classification’. According to your protocol, several included articles should be removed (ICHD diagnosis was not confirmed). This constitutes an important deviation from your protocol.
Results: Data screening and Results: ‘Of the 14 included publications all were cohort studies’. Table 11 (I suppose the number 11 should be replaced with 1) involves ‘A retrospective chart review’, ‘A retrospective crosssectional analysis’ and several other descriptions that do not clearly state the implementation of a cohort design.
Results: Data screening and Results: Your results (Table 11) are not presented in a reader-friendly way. The study characteristics should be presented in one Table. A second Table should be used for the presentation of the individual study findings. Presentation of the relevant findings per outcome (compared to per study) will be probably more reader-friendly. Furthermore, time and region specific trends were not delineated. Authors should search the corresponding (time and region-specific) guidelines and comment on the adherence of each study to them, regardless of the individual study reporting (since several articles did not comment on that). Region and time trend were apparent based on Table 1, but your conclusions are inconsistent with these findings. Results lack a presentation regarding some of the study questions. For example, no mention is found in the whole section regarding the factors that influenced preferred treatment of migraine in the ED (e.g., Friedman 2014 presented several interested associations that are not commented on at all, Young 2017 found important associations as well) or the factors that precluded adherence to evidence-based migraine guidelines (only some examples are given in the section Discussion). Also, study specific issues came to my attention
e.g., Table 11: Gunasekera, 2020: ‘Approximately 46.4% (n= 325) patients with a discharge diagnosis of migraine were treated with opioids, making it the most used medication to treat acute migraines’ and ‘60% (n=451) of patients were given antiemetics (e.g. metoclopramide, ondansetron and prochlorperazine) 51.8%(n=385) received paracetamol’ come in contradiction with one another.
Results: Data screening and Results: ‘Interestingly none of the studies mentioned concepts such as sustained pain free response or 2-hour pain freedom as a treatment goal’: readers may not be familiar with the IHS guidelines for controlled trials in the management of migraine. You should not use such specific concepts without providing their origin (references) and full background (other potential outcomes of interest, etc).
Results: Data screening and Results: ‘Shao et al. (2017) illustrates the lack of geographic trends in their observations of opiate prescribing patterns in three hospitals in Connecticut USA, where they found that the rate of prescription of opiates varied from 6.9% to 69.9%.’: since there is such a large variability in 3 hospitals (7-70%) how come there is no geographic trend? Also, the said study is entitled 'The presenting and prescribing patterns of migraine in an Australian emergency department: A descriptive exploratory study', not in USA. I suppose that instead of Shao et al. you referred to Young et al. who reported that opioid use was much more common in community hospital compared to academic hospital settings. However, contrary to the authors, you report the absence of variability, while you miss the reported association patterns.
Results: Data screening and Results: ‘the studies extracted here showed no discernible geographic trend in opioid prescriptions.’: again, your findings are inconsistent: Table 11: Supapol, 2013: ‘In terms of adherence to standard therapy 6 of the 12 hospitals did not deviate significantly from the standard (0%-8% opiate prescription). The other 6 had proportions ranging from 24% to 69% which was not consistent with recommended treatment.’ The high opioid use for migraine in ED is revealed by most studies, but significant variability is apparent. Variability between 7% and 70% (see previous comment), as well as lower than 8% and higher than 24% variability should be considered important. Reasons for this variability are probably reported (or at least speculated) by the authors of each study.
Above are some examples of the inaccuracies and inconsistencies I detected. Of course, there are probably additional issues ought to be addressed.
Results: Data screening and Results: ‘Friedman et al (2014) also agreed the with this sentiment’, please reformulate.
Discussion: 'Future observational research into this area should focus on characterizing the extent which prescription patterns deviate from recommended guidelines.' Considering that all the relevant articles are retrieved, you could provide this information.
Author Response
Thank you for your constructive comments and advice. We appreciate the time and effort you put into them and have revised the manuscript as per your suggestions.
Abstract All references have been removed from the Abstract
Introduction Revisions to the 1st Paragraph have been actioned as per your recommendations as follows:
“Migraine headaches are a common, debilitating, and costly neurological disorder. Migraine headaches affect up to 16.6% of the general population [5]”
Revisions to the 2nd Paragraph have been actioned as per your recommendations as follows:
“Migraine sufferers often present to the emergency department (ED) seeking relief from their symptoms, with data from the United states showing at least 1.2 million presentations to the emergency department every year [3]. When compared with other health services EDs received almost 20% of migraine presentations[13]”
The last paragraph has been removed as per your recommendation.
Experimental Section Types of studies: This section has been clarified as per your recommendations as follows:
“The types of studies considered for this review included case control studies , cohort studies , randomized controlled trials, systematic reviews and meta-analyses. All peer reviewed experimental, descriptive and observational studies reporting quantitative data were considered”
Supplemental material has been included as part of the revised manuscript.
Types of studies: This section was written to encompass a variety of qualitative and quantitative methods. This section has been clarified as per your recommendations.
“The types of studies considered for this review included case control studies , cohort studies , randomized controlled trials, systematic reviews and meta-analyses. All peer reviewed experimental, descriptive and observational studies reporting quantitative data were considered.”
Methods: In the original manuscript we provided (i) a reference to the Arksey and O’Malley methodological framework (ii) a description of specific details relevant to our scoping review (iii) they keywords used to search databases.
We recognize the potential overlap between databases; however, all of the above databases were searched to ensure the breadth of our search included non-overlapping publications. For a description and example of the search strategy used please refer to Supplemental material S1 and Table S1 respectively. Experimental Section: Screening and Data Extraction: This section has been modified as follows: “The web-based tool Covidence was used to aid the process of removing duplications, screening, and data extraction.” Results
The number of studies retrieved from each individual database is now included in Table S3.
Results: Data screening and Results: Methodological information has been moved as per your recommendation.
Regarding your comment on the inclusion deviating from protocol. We realized that there was an unintentional omission of the ICD criteria from our inclusion criteria. This has now been corrected. We do however stress that our inclusion criteria for migraine diagnoses are that they must be made by a medical professional or determined to have migraine based on the either the International classification of headache disorders (ICHD) or international classification of diseases (ICD) criteria. We believed this criteria achieved a good balance between inclusivity and accuracy in the diagnosis of migraine headache. The way migrainee was defined was outlined in our extraction table, allowing the reader to make their own judgements on how authours had choosen to define their patient population.
In response to your comment regarding “Table 11” The second '1' is in superscript and refers to the footer of the tables. This formatting is consistent when viewing the pdf downloaded from https://www.preprints.org/manuscript/202012.0745/v1 . We will raise this concern with the editor. We have altered the table to enhance readability in line with your recommendations. Data in our previous “table 1” is now shown in “Table 1 and Table 2”. Note these have 1 in superscript describing abbreviations relevant for the tables.
We thank you for raising the issue of incorrect referencing in our manuscript. We have revised this accordingly. Regarding your comments on Friedman 2014, we would like to reassure you that there was no intentional omission of opinions or trends. Human error had occurred, and Friedman 2014 (a conference abstract) was referenced instead of Freidman 2015 (A paper published reporting initial findings presented in a conference). We have revised and incorporated the views of Freidman 2015 as originally intended.
We believe our revised comments on Friedman 2015 and Young 2017 are appropriate.
In regards to your comments regarding Supapol 2013. We acknowledge that the difference “trend” referred to could be clearer. In this case we are implying that a geographical location is not corelated with the propensity for opiate prescription.
This is a separate issue from very clear over prescription of opiates and underutilization of triptans.
The results section has been revised extensively to more clearly communicate trends observed and comment on variability.
Discussion We have made the reference to prior discussions more clear in the revised manuscript. The further characterization we refer to is described in the following quotes: “Comments made by Young et al. 2017 were also valid, they suggested that individual factors such as previously reported poor response to triptans by patient, physician unfamiliarity with medication and high cost of triptans may also contribute to low use.” “It is believed that a proper characterization of whether opiates were used as first line or rescue therapy is needed to make a complete judgement on the appropriate use of opiates within an institution. It is our opinion that to comply with best practice guidelines opiates an ED should have near zero levels of opiate as first-line therapy for migraine and opiates should not encompass the majority of rescue therapy prescriptions. We encourage future research to consider these details when conducting observational studies for opiate prescriptions in the ED. ” |
Reviewer 2 Report
The paper “An Evaluation of Medication Prescribing Patterns for Acute Migraine In The Emergency Department: A Scoping Review” by Jun Hua Lim 1, Leila Karimi 2 and Tissa Wijeratne, deals with a topic of great importance and intersection which also has great importance and possible repercussions on clinical practice.
Precisely because the topic is important and of great interest, it is necessary that the data analysis and the discussion are clear and methodologically correct. Therefore, the presentation structure must also be clear and easily legible. This aspect was also recalled by the Authors in the limitation section where they say : “to gain a global perspective on the issue is further hindered by only reviewing literature published in English. While a global may not have been achieved it is felt that the narrative that we provided here would be beneficial and translatable to many other contexts.”
So far, the main concerns are:
In many sites is reported the Intravenously use of triptans: this is incorrect. This type of adminisstration was used in 1990’s in the early trials, but this way of administration was immediately suspended. In fact, Sumatriptan should not be given intravenously, because of its potential to cause vasospasm.
Page 16: “Wasay et al. 2006 (based in Pakistan) and Minen et al. 2020 (study based in the New York USA) reported that they had no access to intravenous trip-tan medication. It is important that EDs and hospital pharmacies stock IV triptans and dihydroergotamines to allow compliance with evidence-based medicines.”. This sentence should be corrected: the IV use is use advised against or prohibited, in US; EU and other countries.
Table 11: this table is completely unreadable and difficult to understand. On my opinion you should transform the items and information in aggregate information divided in more tables, possibly classified for papers or items.
I agree with your conclusion that “88% stated that their practices were non-adherent to international guidelines” and “that there is a worrying tendency for the over prescription of opiates and under prescription of triptans. We observed a significant evidence-practice gap in the management of acute migraine in ED. … Ongoing advocacy, educational programs and translational research into this area should focus on addressing this issue as a matter of high priority.”
Author Response
Thank you for your comments. The oversight regarding the route of administration has been addressed in our revision.
The changes your recommended for page 16 have been implemented as follows:
We have altered the table to enhance readability. Data in our previous “table 1” is now shown in “Table 1 and Table 2” and the study characteristics are now separated from the key findings.
We appreciate your constructive comments and advice. We hope you would support the revised submission as well.
Reviewer 3 Report
important stduy which demonstrates that still many patients with migraine receive inadequate treatment
Author Response
We appreciate your comments. We hope you will support the revised submission.
Reviewer 4 Report
This topic has been looked at over the years as cited. Never-the-less it is still relevant that over 2 decades we have not moved forward.
The table was very difficult to read in the current format. Perhaps the table could be redone with the most pertinent information so that the data can be viewed easily and the more comprehensive version provided as an appendix.
It would be useful to clarify which publications were collated by medical or neurological authors and which by ED departments to see whether there is any difference in conclusion. Relevant to this it would also be worthwhile looking at ED recommendations for management of headache in the countries where Neurological guidance exists; this will mainly be in the US given the number of publications from the US; thus what are the American Headache Society guidelines and what are those for the US ED services. This may give some insight as to where the problem may be.
The sumatriptan preparation used and recommended parenterally is usually subcutaneous and not intravenous. Thus the authors may wish to recheck this.
Author Response
We appreciate your constructive comments and advice. We hope you would support the revised submission as well.
We have altered the table to enhance readability. Data in our previous “table 1” is now shown in “Table 1 and Table 2” and the study characteristics are now separated from the key findings.
Indeed, your thought of outlining medical and neurological authors may produce a difference in conclusion. This is certainly something that will be considered if a systematic review was to be preformed following this scoping review.
The oversight regarding the route of administration has been addressed in our revision.
Round 2
Reviewer 1 Report
I have no other comments
Author Response
We appreciate your constructive comments and advice.
All the suggested revisions were made.
In addition to the revisions made, we updated table 1 and changed the column heading in table 1 from “country and context” to “country and setting”, as we believe this more accurately reflects the contents of the column.
We hope that the revised version is to your liking and thank you for your efforts in reviewing our work.
Reviewer 2 Report
In the present form our recommendations have been accepted . The table 2 is still heavy to read, but the paper has been improved.
Author Response
We appreciate your feedback. We have revised the manuscript again and improved the readability of both tables.
We hope that you will support the revised manuscript as well.